# Climate Refuges in Nigeria for Oil Palm in Response to Future Climate and Fusarium Wilt Stresses

**DOI:** 10.3390/plants12040764

**Published:** 2023-02-08

**Authors:** Robert Russell Monteith Paterson, Nnamdi Ifechukwude Chidi

**Affiliations:** 1CEB—Centre of Biological Engineering, University of Minho, Gualtar Campus, 4710-057 Braga, Portugal; 2Department of Plant Protection, Faculty of Agriculture, Universiti Putra Malaysia (UPM), Serdang 43400, Selangor, Malaysia; 3Plant Pathology Division, Nigerian Institute for Oil Palm Research, Benin City PMB 1030, Nigeria

**Keywords:** longitudinal, latitudinal trends, CLIMEX, *Elaeis guineensis*, *Fusarium*, CSIRO, MIROC

## Abstract

The detrimental stresses of future climate change are well known and decisions are required to reduce their effects. Climate and disease stresses cause severe damage to plants and it is essential to understand how they will respond. Oil palm (OP) is an Fusarium important crop for many countries. The palm originated in Africa, where palm oil is produced in the largest amount within the continent by Nigeria. OP becomes stressed by climate change and wilt, a devastating disease of OP in Africa. Previous methods to determine the suitability of future climate on OP in continents and whole countries were applied to Nigeria, which is the first time an individual country has been assessed in this manner. Climate maps of Nigeria were divided equally into 16 regions from north to south and east to west to determine the future suitable climate for growing OP. CLIMEX and narrative modelling were used to determine suitability for growing OP and Fusarium wilt incidence for current time and 2050. Maps from published papers were employed directly thereby facilitating the procedure. A distinct latitudinal increasing trend from north to south in suitable climate was observed, which was unexpected. A decreasing longitudinal trend from west to east was also observed. These differences in suitable climates may allow refuges for OP in the future. The growth of OP in the south of Nigeria may be largely unaffected by climate change by 2050, unlike the north. The procedures allow policy decisions at state and national levels to be made from empirical data, which do not otherwise exist. States with low amounts of OP and where the climate deteriorates greatly, could usefully be abandoned. Other low palm oil producers, where the climate does not deteriorate greatly, could be encouraged to develop OP. Little requires to be done in the high producing states where the climate does not deteriorate. In all cases, the environmental impacts require thorough assessment. Climate change requires reduction as indicated in recent Conference of the Parties meetings.

## 1. Introduction

Palms are important plants within Africa. Oil palm (OP) is the most economically important palm, but is associated with environmental problems and health-related issues [1,2]. The palm produces palm oil which is included in 60% of supermarket products, such as foodstuff and cosmetics. It is also used as biodiesel and for domestic cooking.

Most palm oil is produced in Southeast Asia, particularly Indonesia and Malaysia. However, the OP originated in Africa, as the Latin name, *Elaeis guineensis* indicates. There is a significant palm oil industry in Africa, where Nigeria is the largest producer [3]. Akwa Ibom state produces the most palm oil and has the largest plantation in Nigeria, whereas Edo has one of the largest plantations [4]. Edo represents high volume production for Nigeria, with over 118,264 hectares of land given over to plantations and the government of Edo has allocated over 180,000 hectares for novel plantations. Imo state supports overall Nigerian capacity and has the biggest plantation, producing 150 tons of palm oil daily. Cross River state has over 360,000 hectares under cultivation, representing one of the largest areas and the Cross Rivers government cultivates 1,000,000 palm estates, producing 5,000,000 tons of oil annually. Ondo state is also one of the largest producers, and Okitipupa Oil Palm plc, a well-known processing company, is located in the south of the state. Other high-producing states are Delta, Enugu, Bayelsa, Abia and Rivers [4]. Nigeria and the states will be affected by climate change and OP production will be threatened [5], which will require implementing amelioration procedures through new policies [6].

The highly detrimental impact of climate change is now almost universally accepted. It has been, of course, been recognized recently at the Conference of the Parties (COP) 26 meetings in Glasgow, Scotland [7]. The issue is taken extremely seriously by governments, scientists and the general public, and requiring policies to combat the effects.

Climate change threatens crop production [8]; however, the effects on tropical crops have not been well investigated in Africa [5]. OP is a tropical crop where expansion often occurs at the expense of forests [9,10,11,12,13,14]: a major potential achievement of COP 26 was a declaration on ending deforestation to reduce climate change [15]. Dislich et al. [16] determined that ecosystem functions decreased dramatically upon the introduction of OP plantations and climate change effects on OP will increase economic and social problems in producing regions. For example, deforestation-related procedures are associated with zoonotic human diseases relevant to pandemics [2]. Climate change requires prediction to mitigate its effects [17], and CLIMEX modelling is essential in understanding the impacts on species distributions [18]. In addition, future climate may allow novel agricultural areas to become available and benefit crops in some cases. Sloat et al. [19] referred to in situ adaptation of crops involving the action of humans, in contrast to new geographical distributions of crops. In situ adaption to climate change has also been studied. How crops adapt by moving to novel areas needs better understanding as a response to stress caused by climate.

Latitudinal movements in climate ranges of plants are known [18], although these are unlikely in tropical regions as distances to refuges are prohibitively large. Nevertheless, OP may adapt to stress by growing in refuges where the stress factors are reduced, which may occur with palms more generally. OP is grown currently in optimal climatic conditions [20,21]. Interestingly, longitudinal movement of suitable future climate for OP was suggested for South America, Southeast Asia and Africa [22], indicating refuges for growing OP. Furthermore, Paterson [23] considered future Fusarium wilt disease of OP in three African countries, including Nigeria. It may be useful to assess the situation within countries, rather than for whole countries or continents, to determine information at the local level.

In the current report, climate maps of Nigeria were used to determine the latitudinal and longitudinal effect of climate change on suitability for growing OP in Nigeria, useful for crop management and policy purposes.

## 2. Results

### 2.1. Combined Suitable Climate

Figure 1 indicates the latitudinal trend of combined suitable climate (CSC) for 2050 over the various regions from north to south (i.e., north-north to north, north to south and south to south-south). Currently, an increase in CSC has been observed from ca. 40% to 100% from north-north to north, and then 100% was maintained in the south-south. There was predicted a large decrease in CSC in the north-north and north zones and a small decrease in the south zone by 2050. The CSC in the south-south zone is predicted to remain at 100% by 2050. There were large decreases in the north-north and north zones, with small decreases in the south zone when Δ CSC was considered. The south-south zone was unchanged. The trend line showed an initial decrease of 50% CSC in the north-north zone to −4% in the south-south zone (Figure 1), a difference of 46%. The smallest decrease in CSC is in the south-south at 0%, and the largest decrease is in the north zone at 52%.

**Figure 1 plants-12-00764-f001:**
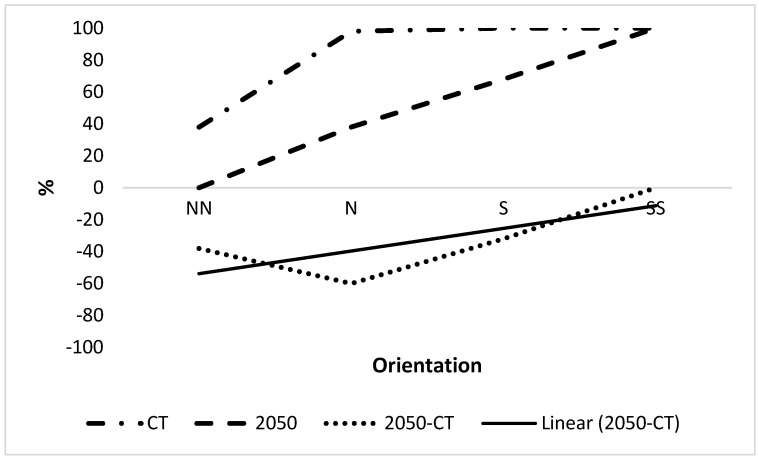
Latitudinal trend of combined suitable climate. Orientation abbreviations are NN, N, S and SS which represent north-north, north, south and south-south respectively of Nigeria excluding parts with unsuitable climate for growing OP as indicated in Figure 2. CT is current time.

**Figure 2 plants-12-00764-f002:**
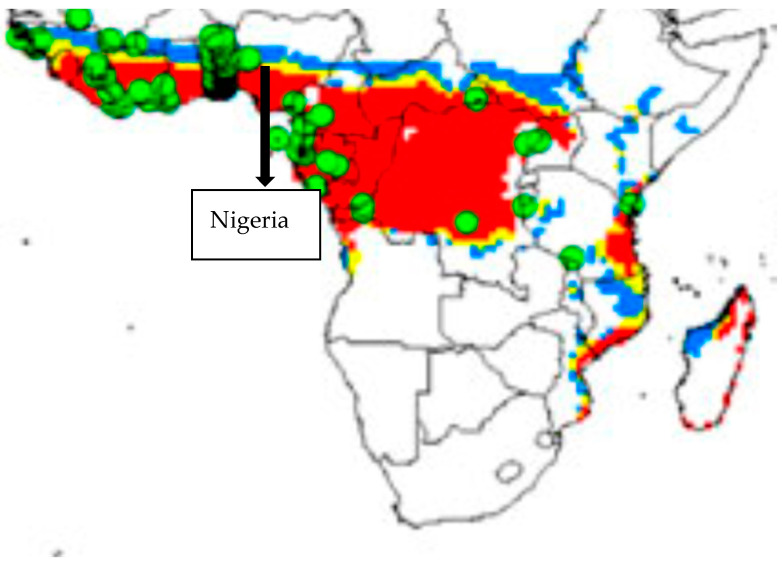
Suitable climate map for OP growing in Africa adapted from [24] with Nigeria indicated. Highly suitable climate (red), suitable climate (yellow), marginal climate (blue) and unsuitable climate (white) in current time. Green spots are existing OP plantations. The map was divided equally into 16 regions from north to south and west to east to determine the climate suitability of each region. The unsuitable climate region in current time was not included in all determinations.

The longitudinal trend in CSC is provided in Figure 3. There is a high level of CSC from west-west to east-east currently, which decreases in each case in 2050 by ca. 20%. Δ CSC is ca. −20% in west-west, which decreases slightly in the east-east. The smallest decrease in CSC is in the west-west at 16%, and largest decrease is in the east at 23% (Figure 3. The trend line for Δ CSC indicates a decrease from ca. 21% in the west-west to 30% in the east-east; hence, there is a moderate decreasing trend from west to east.

The north zone had the largest decrease in CSC of the latitudinal zones (Figure 1) by 2050. This is illustrated in Figure 4. There was no change in CSC in the west-west zone by 2050 and then a greater decrease until the east zone at −75% CSC, which then slightly increased in east-east to −72%. Hence, there was a highly pronounced decreasing longitudinal trend in CSC from west to east in the north sector.

The east zone had the most significant decrease in CSC of the longitudinal zones (Figure 3). It was considered in terms of the four latitudinal regions within it (Figure 5). CSC was 0% in the north-north zone by 2050. However, the largest decrease was in the north zone. A small decrease was observed in the south zone, and a high amount of CSC remained. ΔCSC decreased dramatically to the north zone from the north-north zone and increased in the south and then further increased the south-south zone.

### 2.2. OP Mortality and Fusarium Wilt

Figure 6 indicates the change in OP mortality and acute and chronic Fusarium wilt from the change in CSC for OP for 15 regions. The 16th region had zero CSC and is predicted to remain at zero in 2050, and is not presented in Figure 6. In general, the northern regions have the greatest mortality and disease incidence compared to the southern regions. There was a small level of mortality and more moderate disease in the north/west-west. South-south/west-west, south-south/west, south-south/east and south-south/east-east had the lowest levels of disease, which were at the base line incidence rate. There was no mortality from climate change in the country’s extreme south. North-north/west, north-north/east, north-north/west-west and north/east-east had 100% mortality from a change in climate under this scenario and no disease. The highest mortalities and disease in the southern regions were for south/east-east, south/east, and south/west-west.

## 3. Discussion

The maps in Figure 2 and Appendix A are presented “as used in the study” and taken directly from reference [24]. They have not been enhanced on purpose as this improves the utility of the method, in that maps can be taken directly from publications available on-line. This approach has been used in other OP papers [5,22] which set a precedent. The concept is discussed by Paterson [25]. The procedure can readily be applied to other papers where maps are available for, e.g., other plants.

Plants may respond to stress by growing in refuges where it is reduced. In Nigeria, there was a large increasing latitudinal trend in CSC from north to south Nigeria, consistent with a trend to the South Pole due to climate change (Figure 1 and Figure 5), leading to potential refuges. Refuges were considered unlikely in the tropics [18], but is a clear conclusion from the present results and so may be possible. There was a greater latitudinal trend than a longitudinal trend for Nigeria. However, there is a highly significant increasing west-to-east trend in suitable climate for growing OP in large areas of Africa [5] and within specific zones within Nigeria (see results).

Table 1 indicates the CSC change for Nigeria’s regions and states by 2050. The 100%, very high, high and medium/high decreases are in regions with a low level of OP agronomy. Hence, there is no scope for increasing production from these regions, and the OP industry in these regions will become unsustainable by 2050. Consequently, there may be little advantage in advocating policies to encourage OP production in these states. Kwara state is included in the north-north/west-west region which had a total loss of suitable climate. Part of Kwara is in the north/west-west region, with only a small reduction in CSC, so that OP production may be possible in that part of Kwara. Policies that encourage protection of OP production as outlined by Paterson [6] could usefully be implemented in this state. In the north/west region, Ekiti is a low-producing state parts of which have a high to a medium reduction in CSC and may become unsustainable. The state is also present in the north/west-west, which had a small decrease in CSC and so OP production may continue. Efforts to protect OP from climate change in the north/west-west, by the methods in reference [6], needs consideration, although production in the other region could perhaps be abandoned. Kogi state had a high and high/medium reduction in CSC and is currently only a low palm oil producer so this state may become unsustainable and could perhaps be abandoned in the future and would not affect overall production of palm oil in Nigeria a great deal. A medium level of reduced CSC was observed in south/east-east and south/east, which contains a significant part of the high-producing state of Cross Rivers and the moderate-producing state of Enugu. Considerable efforts should be made to ameliorate the effects of climate change on OP in this state by, for example, implementing policies that advocate using the methods outlined in Paterson [6]. Cross Rivers is also part of the south-south/east-east region with no CSC reduction by 2050 and so the region would continue to produce palm oil with minimal extra effort and fewer policies would be required to assist the industry. Enugu state would probably not support increased OP production but well considered policies will be required to maintain production (see above).

Ebonyi state will be greatly compromised by 2050. As this is a low-production state, the effect on overall palm oil production in Nigeria will be minimal. It may be undesirable to maintain production here and concentrate efforts in other states. The north/west-west region contains low producing Oyo and Osun states, although they had a small decrease in CSC; consequently, they so may not be badly affected by climate change. Kwara and Ekiti states are also in the north/west-west region, and these parts of the states may not be badly affected. Policies to ameliorate the effect of climate change [6] would be appropriate to maintain these regions. However, the situation in north-north/west-west and north/west regions that include these states may not benefit from putting into action amelioration policies. There was a very small decrease in CSC for the south/west region, which contain Edo and Anambra states. Edo is a high OP growing area and will likely maintain that status until 2050. These regions may not require a great deal of additional resources to maintain production.

Anambra is a low-producing state and will be unaffected by the change in climate by 2050. Hence, policies should be introduced to investigate increasing production in Anambra based on this information. It is absolutely essential that the environmental issues relating to deforestation and biodiversity loss are fully integrated into policies that consider increased production. The south-south/west-west, south-south/west, south-south/east, south-south/east and south-south/east-east regions maintained 100% CSC and the sustainability of OP will also be maintained by 2050. The required policies in these regions would be to assist in maintaining the high production. The moderate-producing states such as Delta, Bayelsa, Rivers and Abia may consider growing more OP to compensate for the losses in the more northly states. The north-north/east-east region is somewhat atypical as it had 0% CSC at the current time. This region contains Taraba and Adamawa, which are low-production states, consistent with the north-north/east-east region containing a large amount of marginal climate for growing OP in current time, which may be sufficient for the low production recorded. However, by 2050 this will be greatly reduced, and, hence, even this low production will be threatened. As mentioned previously, it is essential that all environmental issues are fully considered when OP production is increased.

The methods employed herein allow an optimized and systematic procedure for assessing the suitable climate for OP from climate maps, and avoids using the random longitudinal values obtained when employing countries or only considering longitudinal trends as in reference [5]. The procedure described in the current paper allows longitudinal and latitudinal considerations for regions and a more accurate assessment of the effect of a changing climate. The methods also consider the situation within a single country which is novel and useful for crop management in the country as only one nation is being considered with, for example, a single government determining relevant policies, rather than numerous governments being involved if many countries were being considered. Interestingly, Olivares et al. [26] employed similar databases from 2015 to the current paper to produce climate maps for Fusarium wilt of banana in the individual country of Venezuela. Maps were not produced for future climate and so the data are not comparable. Considering individual countries will be useful for national planning and policy purposes in terms of adaption to climate change stress (see above).

The use of these models based on climatic conditions allows for estimating the change in suitable climate for growing oil palm and the geographic distribution of phytopathogenic agents. They indicate how climatic factors can interact in future climate scenarios associated with climate change [5,22,23,26]. Consequently, the results presented here can be very useful for the design of new strategies for the efficient use of procedures to protect OP from climate change because they are adapted to the geographical area.

The data presented herein will be available for checking in the near future with what actually occurs, and it will be possible to assess more clearly if the data accurately depict what actually occurs. The information in the current paper is based on a “no change” situation and climate change may be better or worse than is predicted currently, affecting OP correspondingly. A very large effort to reduce climate change may occur in the near future, although time is limited for these efforts to be effective. Amelioration of the effects of climate change on OP may be possible [6].

The method allows a more logical procedure for determining the effect of longitude and latitude on suitable climate for growing OP. The possibilities of refuges for growing OP require that the negative environmental consequences, such as deforestation, are fully considered before new plantations are created and must be a priority in managing the novel environmental conditions.

## 4. Materials and Methods

The methods were similar to those in reference [5] and are reproduced here briefly.

### 4.1. Suitability of Future Climate for Growing OP in Nigeria

The Global Biodiversity Information Facility (GBIF) (http://www.gbif.org/, accessed on 9 November 2015) and additional literature on the species in CAB Direct (http://www.cabdirect.org/web/about.html, accessed on October 2015), formed the basis for the collection of data on *E. guineensis* distribution. While the GBIF database returned 2851 OP records, 386 lacked the necessary geographic coordinates and were removed. Therefore, 2465 records were utilized in fitting the parameters. These records are geographically representative of the known distribution of the species [24].

For a comparison of the ability of mechanistic and correlative bioclimatic modelling methodologies, CliMond 10′ gridded climate data [24] were employed for the different methods, guaranteeing uniformity of data relating to climatic factors. Parameters of climate incorporated in the meteorological database are the mean monthly temperature maxima and minima (Tmax and Tmin), mean monthly precipitation level (Ptotal) and relative humidity at 09:00 h (RH09:00) and 15:00 h (RH15:00). The same parameters were also used to project the possible future climates. CSIRO-Mk3.0 and MIROC-H GCM global climate models (GCMs) were used and in conjunction with the A2 Special Report on Emissions Scenarios (SRES) scenario, to model potential future distribution of OP [24].

CLIMEX, a mechanistic niche model CLIMEX software supports ecological research incorporating the modelling of species’ potential distributions under differing climate scenarios and assumes that climate is the paramount determining factor of plant and poikilothermal animal distributions [24]. Species growth potential in the favorable season is denoted by the Annual Growth Index (GIA), while the impact of population reduction during an unfavorable season is established by the cold, hot, wet and dry Stress Indices and their interactions. The Ecoclimatic Index (EI), the product of the GIA and Stress Indices, rates the level of suitability for species’ occupation of a particular location or year. The EI is, thus, an annual average index derived from weekly data of the growth and stress indices of suitability levels of climatic factors denoted by a value of 0 to 100. A species may be established where EI > 0. The current research used CLIMEX to model present and future distributions of *E. guineensis*. CLIMEX output categorized areas according to high suitability, suitability and marginal suitability based on other studies through CLIMEX (5). In the present study, temperature index (DV0 = Limiting low temperature, DV1 = Lower optimal temperature, DV2 = Upper optimal temperature, DV3 = Limiting high temperature); moisture index (SM0 = Limiting low soil moisture, SM1 = Lower optimal soil moisture, SM2 = Upper optimal soil moisture, SM3 = Limiting high soil moisture); cold stress (TTCS = Cold stress temperature threshold, THCS = Cold stress temperature rate, DTCS =Minimum degree-day cold stress threshold, DHCS = Degree-day cold stress rate); heat stress (TTHS = Heat stress temperature threshold, THHS = Heat stress temperature rate); dry stress (SMDS = Dry stress threshold, HDS = Dry stress rate); wet stress (SMWS = Wet stress threshold, HWS = Wet stress rate) and Degree-day threshold (PDD) were fitted according to global distribution data, iteratively adjusted to achieve satisfactory agreement between known and projected species’ distributions globally. See (5) for CLIMEX parameter values used in OP modelling. CLIMEX models provide scenarios and are not accurate predictions of the future climate situation, as with other models.

### 4.2. Narrative Model by Examination of Maps in Granular Detail

Paterson et al. [24] employed the CLIMEX model to provide maps of suitable future climate for growing OP in Nigeria for the current time (CT) (Figure 2) and 2050. A map of Nigeria containing the indications of suitable climate for growing OP (Figure 2) and copied from reference [24], was divided into ten equal sectors from north to south and east to west to determine the latitudinal and longitudinal effect of climate change. The maps permitted the determination of the percentage of the climate categories for each zone within Nigeria. Percentages of highly suitable and suitable climates were determined visually from the red and yellow areas in each map and combined to give the CSC parameter: red represented the highly suitable climate and yellow the suitable climate for growing OP. The average values obtained for CSIRO and MIROCH maps were employed as the final CSC values. The change in CSC from future climate was determined by subtracting the CSC values from 2050 from those of CT. This indicated the effect of climate change on the suitable climate for OP growth. The map in Figure 6 which indicates the location of Nigeria which is similar to maps used in previous publications [5,22]. The part representing Nigeria was divided into 16 sectors running north to south and east to west. An approximate working map is available in Appendix A to indicate the divisions used within Nigeria. The percentage CSC was determined for the land in each sector and plotted. The presentation of the actual working document is important to illustrate the exact method used rather than presenting a more stylized version of the map. The trend line was added using Excel software. The change in CSC from CT to 2050, as discussed in the results and the discussion, is abbreviated to “Δ” for convenience. To determine OP mortality and Fusarium wilt, a 12% reduction in CSC was assessed to correspond to 5% OP mortality, 50% increased acute wilt and a 10% increase in chronic wilt by 2050 [23].

## 5. Conclusions

This paper indicates how oil palm will respond to climate and disease stress by spreading to refuges within Nigeria. Clear longitudinal and latitudinal trends were observed for CSC. The results indicate that latitudinal trends are possible within the tropics, despite this being considered unlikely [18] due to the long distances involved in the potential spread of plants to find refuges (see Introduction) which may relate to the small distances involved. OP may find refuges towards the south and west of Nigeria in response to future climate.

The described procedures allow management decisions regarding future climate based on empirical data; to do so otherwise is difficult or impossible. The scenarios described are on a “no change” situation in future predicted climate change.

## Figures and Tables

**Figure 3 plants-12-00764-f003:**
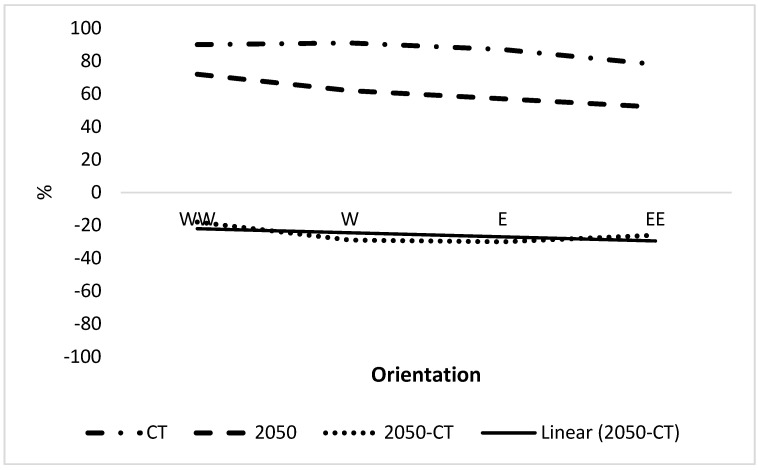
Longitudinal trend of combined suitable climate. WW, W, E and EE represent west-west, west, east and east-east, respectively, of Nigeria excluding parts with unsuitable climate for growing OP (Figure 2). CT is current time.

**Figure 4 plants-12-00764-f004:**
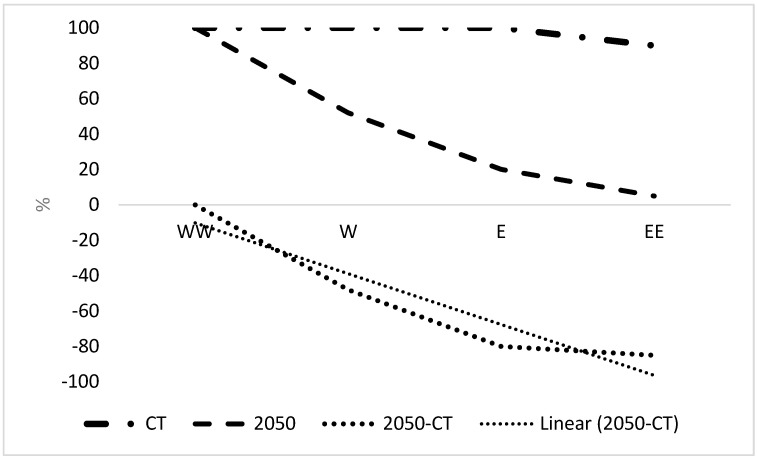
Longitudinal trends in CSC in the north zone of Nigeria. Direction abbreviations are WW, W, E and EE represent west-west, west, east and east-east, respectively. CT is current time.

**Figure 5 plants-12-00764-f005:**
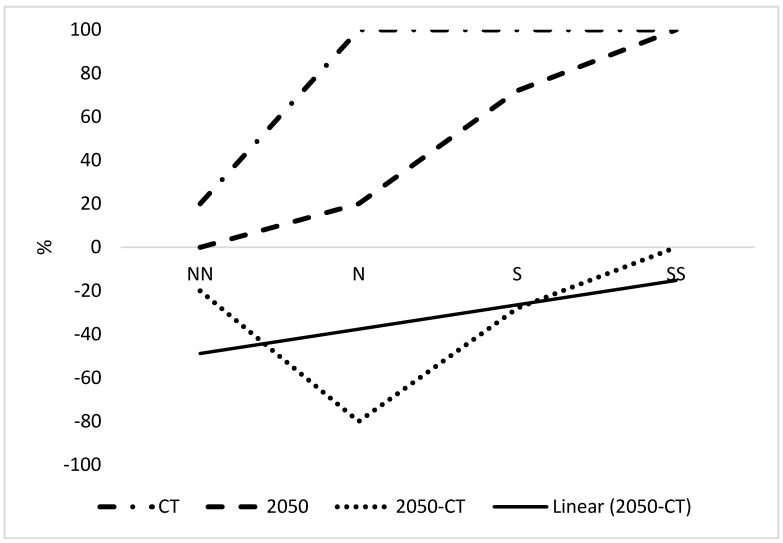
Latitudinal trend in CSC for the east zone of Nigeria. NN, N, S and SS represent north-north, north, south and south-south, respectively, of Nigeria. CT is current time.

**Figure 6 plants-12-00764-f006:**
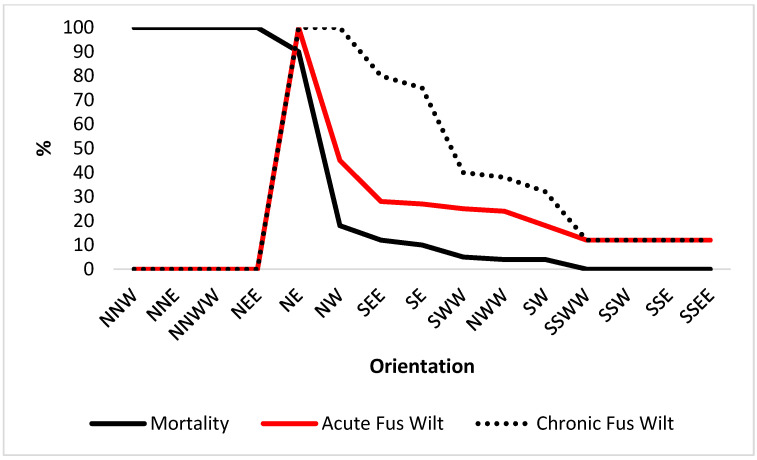
Change in OP mortality, acute Fusarium wilt and chronic Fusarium wilt due to the change in combined suitable climate for growing OP in the 15 regions considered.

**Table 1 plants-12-00764-t001:** Combined suitable climates (CSC) for growing OP in each orientation and associated Nigerian states. The states in bold are present in more than one location and the states with moderate or high palm oil production are indicated in parentheses.

Location	CSC % Current Time	CSC % 2050	Δ CSC %	Decrease in CSC	Nigerian State
North-north/west	65	0	100	Total	Niger, FCT Abuja, Nasarawa, Kaduna
North-north/east	25	0	100	Total	Nasarawa, Plateau, Kaduna
North-north/west-west	55	1	98	Very High	Kwara, Niger
North/east-east	85	8	91	Very High	Benue, Taraba
North/east	100	20	80	High	Kogi, Benue
North/west	100	57	43	High/medium	Ekiti, Kogi
South/east-east	100	72.5	27.5	Medium	Ebonyi, Cross River (High)
South/east	100	80	20	Medium	Enugu (moderate), Ebonyi
South/west-west	100	92.5	7.5	Low	Ogun, Lagos, Ondo (High)
North/west-west	100	95	5	Low	Oyo, Osun, Kwara, Ekiti
South/west	100	97	3	Low	Edo (High), Anambra
South-south/west-west	100	100	0	None	Delta (moderate), Bayelsa (moderate)
South-south/west	100	100	0	None	Imo (High), Rivers (Moderate)
South-south/east	100	100	0	None	Abia (moderate), Akwa-Ibom (Very High)
South-south/east-east	100	100	0	None	Akwa-Ibom (Very High), Cross River (High)
North-north/east-east	0	0	0	Marginal suitable climate is reduced by 2050	Taraba, Adamawa
North-north/west	65	0	100	100% Mortality	Niger, FCT Abuja, Nasarawa, Kaduna
North-north/east	25	0	100	100% Mortality	Nasarawa, Plateau, Kaduna
North-north/west-west	55	1	98	100% Mortality	Kwara, Niger
North/east-east	85	8	91	100% Mortality	Benue, Taraba
North/east	100	20	80	90% Mortality, 100% wilt	Kogi, Benue
North/west	100	57	43	18% Mortality, 100% Acute wilt, 40% Chronic wilt	Ekiti, Kogi
South/east-east	100	72.5	27.5	10% Mortality, 80% Acute wilt, 20% Chronic wilt	Ebonyi, Cross River (High)
South/east	100	80	20	8% Mortality, 75% Acute wilt, 15% Chronic wilt	Enugu (Moderate), Ebonyi
South/west-west	100	92.5	7.5	4% Mortality, 40% Acute wilt, 10% Chronic wilt	Ogun, Lagos, Ondo (High)
North/west-west	100	95	5	3% Mortality, 35% Acute wilt, 7% Chronic wilt	Oyo, Osun, Kwara, Ekiti
South/west	100	97	3	3% Mortality, 32% Acute wilt, 5% Chronic wilt	Edo (High), Anambra
South-south/west-west	100	100	0	0% Mortality, normal wilt levels	Delta (Moderate), Bayelsa (Moderate)
South-south/west	100	100	0	0% Mortality, normal wilt levels	Imo (High), Rivers (Moderate)
South-south/east	100	100	0	0% Mortality, normal wilt levels	Abia (Moderate), Akwa-Ibom (Very High)
South-south/east-east	100	100	0	0% Mortality, normal wilt levels	Akwa-Ibom (Very High), Cross River (High)
North-north/east-east	0	0	0	Large Mortality, Very large wilt levels	Taraba, Adamawa

## Data Availability

Data are contained within the article or Appendix A.

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
