# Peer review of "Climate Refuges in Nigeria for Oil Palm in Response to Future Climate and Fusarium Wilt Stresses"

_plants, 2023, doi:10.3390/plants12040764_

Round 1

Reviewer 1 Report

General comments:

The manuscript should be revised by a native English speaker. As it is now, the written quality is not sufficient for publication.

In the Materials and Methods, the authors describe that data was first collected from two online databases, but the timing of data collection recedes to 2015. As such, a more recent data collection must be implemented in the manuscript, as for sure additional information was made available in those databases in the past 8 years.

The Discussion must be significantly improved to support the findings. Do your results correlate with studies by other authors on other crops/pathogens? What type of policies can be developed based on your results? Could these shifts in OP suitable areas impact native flora/fauna?

References are consistently incorrectly formatted throughout the manuscript. Ex.: line 74, 323: sentences cannot be initiated with the number of the reference; line 83: references should appear in numeric order; Line 119, 279: when referring to the authors of a specific citation, it should appear as Paterson et al. (5), and not as the number of the reference alone; References 31 and 32 are the same

Also, 12 out of 31 references (i.e., almost 40%) are from the authors, and thus, a more diverse bibliography should be incorporated to improve the overall quality and scientific soundness of the manuscript.

Specific comments:

Line 90-92: what were the main findings of the referred works? provide further detail on the outcomes of those studies

Line 124: define CSC in full

Figures 1 and S1 must be significantly improved to allow their proper readability and the software used for their production must be indicated.

Reviewer 2 Report

The work is interesting. This manuscript reports on a study of Future suitable climate refuges in Nigeria for growing oil palm in response to climate and Fusarium wilt stresses. The study design meets the general standards and from what I can judge the data is being collected and analyzed appropriately. This work is an unpublished manuscript with relevant information that should be made public in a scientific journal for discussion among scientists working in the field.

Authors must be careful with spelling (several errors in the text) and writing. However, some comments should be considered before publishing, in this way, the social and scientific relevance of the manuscript would be improved:

Abstract

Line 14: should say: how the organisms

Line 17: add a comma: mortalities, and

Line 21: should say: modeling

Introduction

Line 43: should say: high-volume

Line 44: should say: is given

Line 49: add a comma: 360,000 hectares

Line 50: 1,000,000 palm estates

Line 51: 5,000,000 tons

Line 72: should say: the CLIMEX

Line 79: should say: growing up in

Lin3 97: policy purpose

Figure 1 is of very low quality and inaccurate for publication. I recommend extracting the map of Nigeria from the GIS file to the authors. The figure is a cut from the original image published in doi:10.1017/S0021859616000605.

The new figure must be extracted from the GIS with the legend and scale included.

Results

Line 138: Should say: Fig. 2 the Latitudinal trend

Line 139: add a hyphen: north-north, south-south. The acronyms: nn,n,s,ss must be in capital letters as in the graphic.

In figures 2 to 6, it is not possible to distinguish the lines of CT, 2050, 2050-C, and Linear (2050-CT) in the graph. The figure must be modified, adding color and symbols that distinguish each line in the graph. Specify the meaning of the initials CT,

The x-axis should be at the bottom position of the graph and not in the middle of the graph. Apply these changes to all shapes

Line 144: the largest

Line 148: should say: Figure 3. The longitudinal trend

Line 1498: The acronyms: ww,w,e,ee must be in capital letters as in the graphic. add a hyphen: west-west, east-east. the same for figure 4 and 5

Line 186: delete the repeated word OP

Discussion

add a hyphen: north-north, south-south, etc etc, and make changes throughout the manuscript

Line 208: the overall production

Line 214: not supporting

Table 1 is excessively long; it must be divided into several tables that allow easy reading of the content.

Line 242: an optimized

Line 246: and latitudinal

Line 262: Figures 4 and 5

Line 263: near future to

Line 272: stressful climates

I continue to add a paragraph that summarizes the importance, usefulness, and social relevance, contemporary of the study, specifically pointing out the Impact, Benefit, and Social Projection, something like this (for example):

Line 276: The use of risk estimation models based on climatic conditions allows for estimating the geographic distribution of phytopathogenic agents and their biocontrol agents, as well as determining how climatic factors can interact in both current and future climate scenarios associated with Climate Change (Olivares et al. al. 2021; Navas-Cortez et al. 2007; Castiblanco et al. 2013). Consequently, the results presented here can be very useful for the design of new strategies for the efficient use and greater effectiveness in the activity of well-selected biocontrol agents because they are adapted to the geographical area and crop of action, such as oil palm.

Materials and Methods

Line 299: favorable season

Line 301: unfavorable season

Line 320: OP modeling

Conclusion

Avoid wording that could be construed as unwarranted speculation, tangential issues, or conclusions not supported by your data

Line 347-350. That sentence is not a conclusion of this study. I suggest deleting:  Stakeholders involved in OP need to become involved in Conference of the Parties (COP) meetings, such as the one in Egypt that was held in November 2022 (COP 27) and future such meetings, to ensure that their views are acknowledged at the highest levels of government.

References

In the text, reference numbers should be placed in square brackets [ ], and placed before the punctuation; for example [1], [1–3]

I suggest adding recent references which address the issue in question in Latin American and China territories. Suggested citations are for genuine scientific reasons that emphasize the current topic of study in context:

Navas-Cortés, J.A.; Landa, B.B.; Méndez-Rodríguez, M.A.; Jiménez-Díaz, R.M. Quantitative modeling of the effects of temperature and inoculum density of Fusarium oxysporum f. sp. ciceris races 0 and 5 on the development of Fusarium wilt in chickpea cultivars. Phytopathology 2007, 97: 564 573. https://doi.org/10.1094/PHYTO-97-5-0564   

Olivares B, Rey JC, Lobo D, Navas-Cortés JA, Gómez JA, Landa BB. Fusarium Wilt of Bananas: A Review of Agro-Environmental Factors in the Venezuelan Production System Affecting Its Development. Agronomy 2021, 11(5):986. https://doi.org/10.3390/agronomy11050986

Castiblanco, C.; Etter, A; Aide, TM. Oil palm plantations in Colombia: a model of future expansion. Environmental science & policy 2013, 27:172-83. https://doi.org/10.1016/j.envsci.2013.01.003

Round 2

Reviewer 1 Report

The manuscript has been improved and all my comments have been addressed, thus I find the manuscript suitable for publication.